# Sustainable Green Processing of Grape Pomace Using Micellar Extraction for the Production of Value-Added Hygiene Cosmetics

**DOI:** 10.3390/molecules27082444

**Published:** 2022-04-10

**Authors:** Tomasz Wasilewski, Zofia Hordyjewicz-Baran, Magdalena Zarębska, Natalia Stanek, Ewa Zajszły-Turko, Magdalena Tomaka, Tomasz Bujak, Zofia Nizioł-Łukaszewska

**Affiliations:** 1Department of Industrial Chemistry, Faculty of Chemical Engineering and Commodity Science, Kazimierz Pulaski University of Technology and Humanities in Radom, Chrobrego 27, 26-600 Radom, Poland; tomasz.wasilewski@uthrad.pl; 2COSMEDCHEM, Podleśna 48R, 26-600 Radom, Poland; 3Łukasiewicz Research Network-Institute of Heavy Organic Synthesis “Blachownia”, Energetykow 9, 47-225 Kedzierzyn-Kozle, Poland; magdalena.zarebska@icso.lukasiewicz.gov.pl (M.Z.); natalia.stanek@icso.lukasiewicz.gov.pl (N.S.); ewa.zajszly@icso.lukasiewicz.gov.pl (E.Z.-T.); magdalena.tomaka@icso.lukasiewicz.gov.pl (M.T.); 4Department of Technology of Cosmetic and Pharmaceutical Products, Medical College, University of Information Technology and Management in Rzeszow, Sucharskiego 2, 35-225 Rzeszów, Poland; tbujak@wsiz.edu.pl (T.B.); zniziol@wsiz.edu.pl (Z.N.-Ł.)

**Keywords:** loan chemical extraction, bioactive compounds, cosmetics, grape pomace extracts

## Abstract

This study sought to evaluate the possibility of using grape pomace, a waste material from wine production, for the preparation of cosmetic components. Following the existing clear research trend related to improving the safety of cleansing cosmetics, an attempt was made to determine the possibility of preparing model shower gels based on grape pomace extract. A new method for producing cosmetic components named loan chemical extraction (LCE) was developed and is described for the first time in this paper. In the LCE method, an extraction medium consisting only of the components from the final product was used. Thus, there were no additional substances in the cosmetics developed, and the formulation was significantly enriched with compounds isolated from grape pomace. Samples of the model shower gels produced were evaluated in terms of their basic parameters related to functionality (e.g., foaming properties, rheological characteristics, color) and their effect on the skin. The results obtained showed that the extracts based on waste grape pomace contained a number of valuable cosmetic compounds (e.g., organic acids, phenolic compounds, amino acids and sugars), and the model products basis on them provided colorful and safe natural cosmetics.

## 1. Introduction

Cosmetics, mainly body washes, are amongst the most popular chemical products in the world. They are part of people’s everyday life [1]. In the past, these types of preparations were produced mainly based on soaps. Today, manufacturers of cosmetic products often use synthetic surfactants of petrochemical origin, which can alter the skin flora, causing allergic reactions and skin irritations [1,2,3]. Frequent body washing and continuous use of cosmetics over a prolonged time may have various undesirable effects, which may cause allergic reactions. For that reason, consumers have become more quality conscious towards cosmetics and expect the product they buy to be efficient and safe [4,5].

Increasing hygiene cosmetics’ safety can be implemented in many ways. Particular attention is paid to the manufacture of products based on natural raw materials or natural origin [6,7,8], the use of various types of gentle surfactants [9,10], the introduction of ingredients into products that reduce skin irritation and drying effects [11,12,13], as well as new product forms and technological developments in manufacturing [14,15].

A further important issue is the need to pay special attention to care for the natural environment. This is manifested by actions taken to implement the sustainable development principles [16] or “zero waste” ideology [17]. Scientists, equipped with the most modern research methods and infrastructure, focus their attention on, for example, using production or agricultural waste to produce interesting, valuable raw materials [18]. Such products are effective, inexpensive, and bio-sustainable, and therefore an important alternative to the usual plant extracts more commonly used for the cosmetics industry. An example is grape pomace obtained from wine production [19,20]. Winery wastes are rich in bioactive compounds, edible acids, dietary fibers, etc., and can be used to produce value-added components [21,22].

Various techniques are used to process the waste material. The scientific literature indicates that most attention is paid to various types of extraction process [23]. The most popular method is solid–liquid extraction (SLE), which simply involves solvent application and leaching [24,25,26,27]. Conventional extraction processes often use a lot of solvents and involve a recovery step which is followed by extract concentration via evaporation. Finding the solution to these problems has resulted in the use of novel classes of solvents, such as deep eutectic solvents (DES) [28,29], as well as the development of several unconventional methods in recent years. Such methods include supercritical fluid extraction (SFE) [30,31,32], cloud-point extraction (CPE) [33,34,35,36], microwave-assisted extraction (MAE) [37] and ultrasound-assisted extraction (UAE) [38], as well as pressurized-liquid extraction (PLE) [39,40], and extraction assisted by pulsed electric fields (PEF) [41,42] or enzymes (EAE) [43,44,45]. Some of the improved properties of these unconventional techniques include automation, increased selectivity, higher extraction efficiency, and reduced consumption of extraction solvents [46,47]. Extracts produced from different plants are valuable ingredients for cosmetics, mainly because they contain various biologically active molecules, such as polyphenols [48], fatty acids, minerals, and vitamins. The presence of polyphenolic compounds affects the properties of plant extracts related to their antioxidant, anti-inflammatory, antimicrobial and UV-protective activities [49,50].

Despite the growing popularity of extracts and the development of various extraction techniques for industrial applications, synthesized raw materials are significantly more commonly used. One of the factors that has a significant impact on this practice, is that cosmetic raw materials in the form of extracts, in many cases contain, besides the substances extracted from plant material, various kinds of additive. In the great majority of cases, their presence influences the properties of the preparation produced. Examples include amphiphilic solvents, such as ethanol or propylene glycol. In the case of cosmetics intended for hygiene use, in which sodium chloride is used as a viscosity modifier, amphiphilic solvents make it difficult to obtain the required viscosity of the cosmetic and require a significant increase in electrolyte concentration. As a consequence, the preparations acquire much higher detergent capacity, which in many cases is unfavorable and leads to excessive drying of the skin [10,11]. Therefore, manufacturers often limit the extract content in cosmetics and do not utilize their valuable care properties.

Extraction using unconventional methods, such as SFE, is relatively advantageous, as it has been suggested the methods offer an environmentally friendly alternative approach to extraction by avoiding the use of large amounts of toxic solvents [51,52]. However, the unconventional techniques also require many treatment and sterilization steps between each yield, along with special equipment and materials; hence, the introduction of such extracts into cosmetics is difficult, and often the cost of such extracts is quite high. As a result, relatively low concentrations of such extracts are used in cosmetics [53].

Due to their richness in various bioactive components, extracts from agricultural waste products offer an interesting source of raw materials for the preparation of new cosmetic products. An important issue is the need to develop extraction processes that will be complementary to currently used technologies of cosmetic production, especially in the area of hygienic preparations. An important factor is the reduction of production costs for this type of raw material. Recent directions include the attempt to use various types of waste raw material as a starting material for production of full-value cosmetic raw materials.

Grapes, one of the most commonly consumed fruits in the world, contain a large quantity of phenolic compounds. Therefore, grape berries have been extensively studied due to their bioactive compound content and their potential beneficial effects on human health [54,55,56,57,58]. The reddish-purple pigmentation of grapes, as well as of blueberries, blackberries, blackcurrants, chokeberries, raspberries, and elderberries, is associated with anthocyanins, a common group of compounds that gives color to flowers and fruits. These red- to blue-colored compounds are powerful antioxidants. Even low concentrations of anthocyanins can reduce the amount of UVB radiation interacting with the epidermis, demonstrating the benefits of anthocyanins in preventing UV-induced skin damage [59]. Anthocyanins are also used as natural colorants [60].

Grapes and grape products, as a source of bioactive chemicals applied in cosmetic manufacture, constitute an environmentally friendly alternative for generated residues. The polyphenols from these sources are potentially very valuable to the cosmetics industry, which is always in search of new and sustainable sources for active ingredients [61,62,63].

Considering the limitations of industrial production, as well as the seasonal availability of winery by-products, the extracts from this agroeconomic activity require preservation for further use in valorization applications and processes. Once extracted, bioactive compounds can be stored as dried extracts. The drying technique used needs to be considered to provide products of value to the industry [64].

This paper presents the results of research concerning an innovative approach to cosmetics production. In traditional solutions, extracts are added during the main process of cosmetic mass production. In the proposed solution, it is assumed that the extraction process will be one of the cosmetic production stages. In this approach, the extraction medium is an aqueous solution of compounds that are components of the produced cosmetic. The composition of the extraction medium is chosen in such a way that aggregates (micelles) are obtained in the bulk phase, providing the possibility of efficient leaching of cosmetically valuable components from the plant material, in a similar way to micellar extraction [35,65]. The proposed cosmetic manufacturing method includes a key step referred to as “loan chemical extraction” (LCE), defined as “extraction with ingredients borrowed from the final product”.

The aim of this study was to demonstrate the possibility of using LCE for the isolation of bioactive compounds from waste plant material, specifically, red grape pomace from wine production. Prototypes of natural washing cosmetics (shower gels) were developed and manufactured, and then their parameters related to functionality and safety of use were empirically assessed.

## 2. Results and Discussion

### 2.1. Development of Loan Chemical Extraction for Obtaining Cosmetically Valuable Compounds from Waste Plant Material Using Components Borrowed from the Final Product

The starting idea was to develop an effective extraction medium containing only the ingredients planned for use in the final product. It was assumed that the cosmetically valuable substances contained in the waste material (grape pomace) would be both hydrophobic and hydrophilic. From the final cosmetic composition, the surfactant (decyl glucoside) was selected, which forms associative aggregates (micelles) in aqueous solution. Inside such aggregates, hydrophobic substances can be solubilized. In this regard, this type of system is used in micellar extraction processes [65].

A non-ionic surfactant, decyl glucoside, at a concentration of 2% (*w*/*w*) was used to prepare the micellar extraction medium. To ensure that micelles were present in that concentration of the solution, the critical micelle concentration (CMC) of the surfactant was determined using the surface tension measurement for different concentrations of surfactants. With increasing surfactant concentration, the surface tension initially decreased. At the critical concentration, micelles were formed, and the surface tension reached a plateau (Figure 1). The CMC value of surfactants in aqueous solution was assayed as the concentration corresponding to the breaking point of the surface tension via the concentration curve. Surface tension measurements showed that micellization occurred at a surfactant concentration of 0.005% (*w*/*w*). The 2% (*w*/*w*) surfactant solution used as the extraction medium corresponded to a surfactant concentration 400 times above the CMC.

The micellar solution was characterized in terms of the size of the aggregates formed in the solution and their distribution. Particle size distribution is one of the most important parameters for assessing the stability of colloidal systems.

The obtained particle size distribution is shown in Figure 2. A monomodal particle distribution profile was found for the tested surfactant with an average hydrodynamic diameter of 14 nm, indicating the presence of nanometer-sized entities organized in the form of small micelles.

Nanometer-sized entities are excellent platforms for leaching or stabilizing a variety of active ingredients that can be efficiently delivered to the desired site of action without compromising the activity of the incorporated material. In particular, nanometer-sized entities can be used to deliver conventional natural products that have poor solubility or a short half-life. This is how the composition of the extraction medium with nanometer-sized aggregates (micelles) obtained in the bulk phase, provided the possibility to efficiently leach cosmetically valuable bioactive compounds from the plant material. It was also important that the developed extraction medium, despite the presence of associative aggregates in the bulk phase, had a relatively low viscosity (about 1.25 mPs·s, Hoppler viscometer), favoring the process of leaching active substances from plant material. Since the extraction medium is the component ready to use in the final cosmetic product, the extraction process has been termed “loan chemical extraction” (LCE), in relation to the extraction using ingredients borrowed from the final product. In this study, loan chemical extraction was used to prepare grape pomace extracts for a model shower gel using plant waste generated from wine production. The milled, powdered grape pomace was dispersed in a micellar extraction medium. The obtained extracts, with the same surfactant and preservative content, differed in the amount of grape pomace used in the extraction process; proportions of 10 and 20% (*w*/*w*) of pomace were used to prepare the GPE_10pDG2p and GPE_20pDG2p extracts, respectively. The obtained GPE_10pDG2p and GPE_20pDG2p extracts were used for further investigation.

### 2.2. Determination of Selected Compounds by UPLC-MS/MS

To evaluate the quantity of selected compounds in the grape pomace extracts, GPE_10pDG2p and GPE_20pDG2p, quantitative analysis was conducted by ultra-high-performance liquid chromatography electrospray ionization tandem mass spectrometry (UPLC-ESI-MS/MS). The results, expressed in mg/L, are displayed in Table 1. The examined compounds were divided into four classes: organic acids, phenolic compounds, amino acids, and sugars.

The analysis of the chemical composition of the grape pomace extracts showed that, among the organic acids, tartaric, malic, and maleic were identified and the first two were the most abundant. For tartaric acid, 1104 mg/L and 2817 mg/L were found in GPE_10pDG2p and GPE_20pDG2p, respectively, while, for malic acid, 848 mg/L and 1147 mg/L were found. The content of organic acids in the pomace extracts obtained by LCE was 185 g/kg dry weight (DW), which exceeded the values given by Tinkunova et al. [58], which, depending on the grape variety, ranged from 33 g/kg for Saperavi to 108 g/kg for Rebo. Tartaric acid was present in the extracts in the greatest amount. Tartaric acid has become a common ingredient in skin-care products due to its keratolytic and astringent properties. It moisturizes the skin, stimulates the metabolism, promotes healing and also has an anti-aging effect. The malic acid content in the GPE_10pDG2p was comparable to that reported in the literature [66] and 1.5 times higher in GPE_20pDG2p. The greatest difference in the quantity of organic acids between both tested extract concentrations was for tartaric acid, which in GPE_20pDG2p was 2.6 times higher than in GPE_10pDG2p.

The major phenolic compounds in both the GPE_10pDG2p and GPE_20pDG2p extracts were quinic (4.88 mg/L and 10.86 mg/L, respectively) and gallic acid (1.87 mg/L and 4.91 mg/L, respectively), but a small amount of quercetin was also detected. The amounts of gallic acid extracted from grape pomace reported by other authors [67,68] were lower than that obtained by the loan chemical extraction method. On the other hand, the same authors reported amounts of quercetin comparable to those obtained in our experiment. The aqueous-alcoholic extracts presented by Tinkunova et al. [58] showed larger amounts of both mentioned compounds. In addition, catechin, epicatechin, and catechin 3-gallate were the main flavan-3-ols in grape pomace. In both the analyzed extracts, the first two catechins did not differ in quantity in a statistically significant way (5.76 mg/L for GPE_10pDG2p and 5.36 mg/L for GPE_20pDG2p for catechin, and 4.40 mg/L and 4.64 mg/L, respectively, for epicatechin). This was in contrast to catechin 3-galactan, which was found to be 1.6-fold higher in GPE_20pDG2p. These amounts of catechin and epicatechin were similar to those reported by Tinkunova et al. [58].

Methionine is an amino acid that supports the natural regeneration processes of connective tissue, skin, hair, and nails. Tryptophan is the most important compound for the intradermal synthesis of melatonin, which is a substance with a strong cyto- and tissue-protective effect on many levels and is involved in mechanisms of molecular and cellular damage, both at physiological and pharmacological concentrations [69]. Methionine and tryptophan were found at the same levels in GPE_10pDG2p (2.8 mg/L and 3.1 mg/L, respectively), but in GPE_20pDG2p, the quantity of these essential amino acids was more differentiated (6.1 mg/L and 7.0 mg/L).

Of the soluble sugars identified, xylose was the major compound, but smaller amounts of sucrose were also detected. Xylose was the major detected sugar, which, in GPE_10pDG2p, reached 6.3% and in GPE_20pDG2p 7.8% of the dry pomace matter, which represents a higher level than was reported by Valiente et al. [70] where 1.7% was obtained.

Grape pomace is also a rich source of polyphenolic compounds [67,68,71], and other compounds, such as vitamins [58], lipids, and fatty acids [72], have also been quantified.

### 2.3. Determination of Anthocyanins by UPLC-ESI-MS/MS

Figure 3 and Figure 4 show the profile obtained by UPLC-ESI-MS/MS and individual structures of identified anthocyanins in an exemplary GPE_10pDG2p sample, respectively.

A list of the eleven identified anthocyanins in the tested GPE_10pDG2p and GPE_20pDG2p samples are shown in Table 2 (relative to the acquisition in a positive-ion mode). These compounds were identified and confirmed based on MS^2^ fragmentation of selected mass-to-charge (*m*/*z*) signals. The detected anthocyanins were related to the four different aglycons: cyanidin, petunidin, malvidin, and peonidin. Four types of glucosides, three types of acetyl glucosides, and four types of cumaroyl glucoside derivative were identified. The presence of these compounds in grape by-products has also been confirmed by other authors [67,68,71,73], which makes them a potential biomarker for the consumption of these food products.

Based on a comparison of the peaks, the signals for Pet 3-glc and Mv 3-(6-acglc)-5-gl in GPE_20pDG2p were almost 2 times higher, and for Pet 3-(6-cum)glc, 1.4 times higher, than in the GPE_10pDG2p sample. However, for the majority of the determined anthocyanins, the two-fold increase in the concentration of the grape pomace extract did not significantly (*p* < 0.05) increase the amount of identified anthocyanins.

### 2.4. Total Phenolic, Flavonoid, and Anthocyanin Content

The concentration of total phenolics, total flavonoids, and anthocyanins, referred to as mg/L of grape pomace extract, are presented in Table 3. The results showed that total phenolic content (TPC) was higher in GPE_20pDG2p in comparison to GPE_10pDG2p. The concentration of total phenols in the GPE_20pDG2p was equal to 709.1 mg GAE/L, whereas the quantity of phenols contained in the GPE_10pDG2p was 543.1 mg GAE/L, corresponding to 31.5 and 54.3 mg GAE/g dw, respectively. The authors did not observe a significant effect on increase in the amounts of phenolic compounds (about 20% more) when applying a twice higher concentration of the extract. The TPC values of the grape pomace closely matched the data reported by several other authors who have observed phenolic content levels between 30 and 75 mg/g dw. Another study, which focused on three varieties of grape pomace extracted with 40% of ethanol (Merlot, Tannat, and Cabernet Sauvignon), reported a similar TPC value (30.2–51.0 mg GAE/g). [71]. In addition, our results are in agreement with the values found by Rockenbach et al. [74] for pomace from four red grapes extracted with acidified (0.1% HCl) methanol (32.6–74.8 mg GAE/g). In another study, the total phenolic content of the grape pomace extract was increased by using a higher glycerol concentration (14.2–26.6 mg GAE/g) [75]. TPC values are influenced by extraction techniques. Table 1 shows that about 25% of the extracted phenolic substances had a flavonoid origin. The quantity of total flavonoids was 139.2 and 177.6 mg/L in the GPE_10pDG2p and GPE_20pDG2p, respectively, corresponding to 13.9 and 7.9 mg QE/g dw of the respective concentrations. Pomace from grape was also studied with regard to the content of extractable anthocyanins. The total anthocyanin content (TAC) of GPE_20pDG2p was about two-times higher (395.7 mg Cyd-3-GLU/L) compared with GPE_10pDG2p (236.7 mg Cyd-3-GLU/L), corresponding to 4.39 and 2.63 mg Cyd-3-GLU/g DW, respectively. The amount of extracted anthocyanin compounds was comparable to that determined by Iora et al. [71] for pomace from different grape varieties (Merlot, Tannat, and Cabernet Sauvignon), where extraction with 40% of ethanol yielded from 1246.85 to 2092.93 mg Cyd-3-GLU/100 g. Negro et al. found yields of 5.3–10.3 expressed as mg/g dw of malvidin-3-glucoside in four monovarietal grape pomaces (Negramaro, Malvasia di Lecce, Primitivo, and Cabernet Sauvignon) [67].

### 2.5. Antioxidant Activity (DPPH, ABTS)

The results of antioxidant tests are presented in Table 3. Similar behavior patterns were observed in both tests, regardless of their action mechanisms. The GPE_20pDG2p sample had higher antioxidant capacities than GPE_10pDG2p. The mean values of Trolox equivalent (TE) of 146.8 and 257.4 µM TE/g were obtained using the ABTS method, and of 216.7, 313.6 µM TE/g through the DPPH method, respectively. In another study on red grape pomaces (Regent and Pinot Noir varieties), mean values of 419 and 477 µM TE/g were obtained using the ABTS method, and of 479 and 480 µM TE/g through the DPPH method, respectively [76]. In comparison, Pérez-Jiménez et al. reported antioxidant activity values lower than our results (124.4 µM TE/g ABTS method) for red grapes [77]. Similarly, the value for the Isabel variety was lower (188.02 µM TE/g for DPPH and 193.36 µM TE/g for the ABTS method) than those found in the present study [73]. The comparison of literature on the antioxidant capacity of winemaking by-products is quite challenging because different analytical methods (such as DPPH, CUPRAC, ABTS, FRAP, etc.), a variety of standards and reference units, and, importantly, differing grape materials are used.

### 2.6. Determination of the Color Parameters of Grape Pomace Extracts

Pomace is a winemaking by-product, rich in bioactive substances, such as polyphenols. The red color of grapes is mainly caused by the anthocyanin compounds group which are found in the skins of all varieties of red grapes. To determine the color difference between the tested concentrations of grape pomace extracts, colorimetric tests were performed.

Although both tested extracts showed an orange-red/brown hue, a comparison of the L*, a*, b* values indicated that the concentration of the extracts differentiated their color which is presented in Table 4.

The lightness (L*) values of the two grape pomace extracts in the DG2p extraction medium were inversely proportional to the extracts’ concentration. The increased values of a* and b* in GPE_10pDG2p reflect a simultaneous color shift of the extract to both red and to yellow. The color shade is indicated by means of the hue angle (ho) values which provide an intuitive expression of the extract’s color. GPE_20pDG2p had a redder orange hue compared to its GPE_10pDG2p counterpart which was more orange-red. The chroma (C*) value decreased with increasing extract concentration which was visually perceived as a greater saturation and deep red in GPE_10pDG2p. The high values of the color differences (ΔE) of the extracts relative to the extraction medium suggest their potentially high coloring power. However, the color difference between both extracts differing in concentrations of grape pomace was not noticeable to an inexperienced observer (ΔE = 1.7) but was significant from a statistical point of view.

### 2.7. Application Analysis

#### 2.7.1. Interactions of Model Washing Gels with the Skin

Interactions between cosmetic products and the skin during the washing process are the most common cause of skin irritations. Using washing cosmetics may lead to skin dryness, damage to skin barrier functions, and changes in skin pH. Surfactants, which are the main ingredients (used in the highest concentrations) in the formulations of this type of cosmetics, are responsible for the negative effects of washing cosmetics on the skin [78,79,80]. To determine the effect of the analyzed model body shower gels (SG) on skin condition, the irritating potential (Zein value) and other parameters, such as skin hydration, transepidermal water loss (TEWL), and skin pH, were determined. The results are presented in Figure 5.

The highest irritant potential was observed for the base sample (SG) without the grape pomace extract in the formulation. Based on literature data, the Zein number of this sample, of around 280 mg N/100 mL, means it is classified as moderately irritant to the skin. The implementation of a new technique—loan chemical extraction—into the cosmetic production process, introducing substances leached in the grape pomace into the model shower gel composition, led to a reduction in the value of the analyzed parameters and resulted in obtaining products with a significantly lower irritant potential. The Zein number was approximately 15% and 25% lower for the SG_GPE_10pDG2p and SG_GPE_20pDG2p samples, respectively, compared to the base shower gel. Both samples were classified as moderately irritating but the SG_GPE_20pDG2p sample was on the border of the classification between no irritant and moderate irritant.

The irritating potential of surfactants strongly depends on the type of surfactant [78,79,80,81,82,83,84,85,86]. There are many ways to reduce the irritant potential of surfactants [13,78,79,80,81,82,83,84,85,86,87,88,89,90]. Literature data postulate that natural active agents of cosmetics can improve the safety of use of washing products [13,78,87,88]. It has been shown that some plant extracts added to surfactant solutions decrease their irritant potential. Polyphenols, flavonoids, proteins, and carbohydrates contained in plant extracts can be incorporated into surfactant micelles, which increases their size and stability and reduces the irritating potential [13,78,85,87,88,89,90]. The reduction in the irritating potential by adding the analyzed grape pomace extracts to the model shower gel formulation was most likely the result of the grape’s active ingredients and their interaction with surfactant micelles. More stable and larger size micellar aggregates release a smaller number of monomers into the bulk phase, which reduces the risk of interactions with skin proteins.

Surfactants contained in washing cosmetics affect the pH value of the skin, causing it to change after the washing process. There is usually an increase in skin pH, which is not favorable. Changes in pH, and especially its increase, may disrupt the skin’s barrier function and slow down the reconstruction of the skin microbiome [13,78,79,80,91]. It was shown that the addition of the analyzed extracts to the formula of the body wash gels did not cause significant changes in skin pH (Figure 6). At 1h after applying the SG_GPE_10pDG2p and SG_GPE_10pDG2p samples to the skin, a slight decrease in skin pH was observed, which, after 2 h, returned to the physiological level and reached the value observed for the control (the area of the skin where the samples were not applied). In the case of samples containing grape pomace extracts, a physiological pH value was also observed after 4 and 6 h, which indicated that the addition of the analyzed extracts to the formulation of washing cosmetics prevented changes in skin pH. At 1 h after the base sample was applied to the skin, an increase in skin pH of about 1 pH unit was observed. Significant increases were also recorded after 2 and 4 h. By only 6 h after the sample was applied to the skin, the pH had returned to its physiological state.

The wide range of biological and pharmacological activities of plant raw materials makes it important to determine the specific effect of plant substances on the human skin. Plant extracts play an invaluable role in this respect, as they are able, not only to provide protection, but also to neutralize the already existing negative effects of external factors on the skin.

The effect of the model body wash samples containing plant extracts on basic skin parameters, such as skin hydration and TEWL, was assessed. The test samples contained extracts from 10 and 20% solutions of grape pomace in the extraction medium. The analyses were carried out at four-time intervals—1, 2, 4 and 6 h after product application. The analysis of the obtained results showed the positive effect of the presence of the extract on skin condition. Grape pomace extracts are rich in many substances, such as secondary metabolites, flavonoids, and polyphenols, and primary metabolites, such as proteins and amino acids, which have a positive effect on the skin. It should be remembered that substances with hydroxyl groups in their molecules contained in the extract can form a hydrogen bond with water, thus binding water in the epidermis [91,92,93]. The results are shown in Figure 7A,B.

Analysis of skin moisture referred to a base and expressed in percentages leads to the following conclusions. First, the effect of pure SG was negative, in the sense that a decrease in skin moisture was observed during the experimental period, although the decrease became weaker with time, starting from over 7% (measured after 1 h), becoming less than 2% after 6 h. The situation was different with respect to SG_GPE_10pDG2p and SG_GPE_20pDG2p. For these, at the beginning, i.e., after 1 h, a decrease in skin moisture was observed for both samples, with values of −6% (SG_GPE_10pDG2p) and −5% (SG_GPE_20pDG2p). The result changed with time, with increase in moisture observed with time for both gels containing the extract. In the case of SG_GPE_10pDG2p, the observed values increased to over 2%, while in the case of SG_GPE_20pDG2p, a positive change in skin hydration of 6% after 6 h was noted compared to the control field.

Analysis of the change in TEWL showed that the values observed for SG varied between negative and positive values, oscillating around zero, and reached −3% after 6 h compared to the control field. After 2 and 4 h from the moment of sample application, the change in TEWL became positive, which indicated that the base sample increased the amount of water evaporating from the epidermis. For the SG_GPE_10pDG2p and SG_GPE_20pDG2p samples, the measured values were negative throughout the experiment. For SG_GPE_10pDG2p, the values ranged from about −5% (after 1, 2, and 4 h) to −10% after 6 h. In the case of SG_GPE_20pDG2p, the observed change in TEWL was about −6% after 1, 2 and 4 h, and decreased to about −12% after 6 h relative to the untreated field. The results showed that the addition of extracts to the recipes of the analyzed products significantly reduced the loss of water from the epidermis after the use of the analyzed samples in relation to the control field. To summarize the above two analyses, SG_GPE_20pDG2p exhibited better values for both skin moisture and TEWL changes.

#### 2.7.2. Determination of the Color Parameters of Cosmetics Containing Extracts

Today, people are looking for natural and organic products. Many new natural extracts provide benefits in the form of pro-health effects due to known antioxidants commonly occurring in the plants or fruits, but also due to their natural color that makes them more desirable. Thus, polyphenolics recovered from the grape pomace can find application as a natural colorant [94,95] and may find use as an extract for a cosmetic formulation [63,67,71,94].

In our investigation, the color parameters of the model hygienic cosmetics formulated with the use of GPE_10pDG2p and GPE_20pDG2p were characterized and are presented in Table 5.

The addition of GPE_10pDG2p and GPE_20pDG2p changed the appearance of the base SG cosmetic and the obtained tristimulus values were significantly higher compared to those obtained for the grape pomace extracts used to prepare the corresponding model products of shower gels.

The same trends as for previously characterized extracts, for lightness and for the a* and b* coordinates, were found. A cosmetic with 10% extract was described as lighter, redder, and more yellow than its 20% counterpart. In addition, the hue angle (ho) and chroma (C*) values showed the previously described tendency for the extracts. The exception was significantly higher saturation (32.7) for cosmetics with GPE_10pDG2p in comparison to GPE_20pDG2p (24.3), which made the perceived color more vivid and bright orange. The calculated color difference, ΔE, compared to the base cosmetic, namely SG, showed the good coloring power of both the tested extracts, with a higher value for cosmetics with GPE_10pDG2p. The color difference between the cosmetics with both extracts was clearly evident (ΔE = 3.9).

#### 2.7.3. Rheological Behavior

In addition to active ingredients, cosmetic products include excipients and additives, such as thickening agents, stabilizers, preservatives, colorants, and perfumes. While the active ingredients, such as bioactive compounds, are the main compounds that determine the function of the products, excipients are designed to dissolve the active compound. They regulate the delivery of active ingredients as well as the aesthetic presentation of the product. Thickening agents increase the viscosity of the product by maintaining the right consistency of the cosmetics, which is essential for the distribution of the active ingredients. The rheological behavior of the model shower gels is presented in Figure 8.

The viscosity and shear stress showed a nonlinear trend with an increasing shear rate, indicating non-Newtonian behavior of all the studied model shower gels. It was observed that the viscosity of the investigated samples decreased with increasing value of shear rate, which characterized the pseudoplastic shear-thinning materials. Up to a shear rate equal to 10 s^−1^, SG had the highest viscosity and SG_GPE_20pDG2p the lowest. The reduction in gel viscosity was related to the presence of the active ingredient, being the extract with a significant proportion of bioactive compounds, which affected the gel-forming ability of the thickening agents in the formulation. However, for the higher values of shear rate the samples had close viscosity values.

#### 2.7.4. Foam Ability

The results of an examination of the foaming ability and foam stability of the shower gels are shown in Figure 9. The obtained results demonstrated that the obtained model shower gels were characterized by excellent foaming ability and foam stability. The aqueous solutions of the tested preparations were able to produce foam with a volume of approximately 500 mL. It was found that the addition of extracts did not significantly influence the foaming properties. Furthermore, it was observed that introducing the grape pomace extract did not affect the foam stability.

#### 2.7.5. Microbiological Stability

Microbiological stability was analyzed for the extracts and shower gels. No bacterial colonies, fungi, yeasts or molds were found as a result of the tests. All tested formulations showed the required microbiological stability. Example plates for microbiological stability testing are presented in Figure 10.

#### 2.7.6. Stability

Stability testing of cosmetic products helps to ensure their high quality and safety for consumers. Organoleptic (e.g., appearance, color, odor) and physicochemical parameters (e.g., pH value, viscosity, weight, signs of separation) were checked in the mechanical loading test. No changes were noted for all investigated parameters and the products were found to be stable.

## 3. Materials and Methods

### 3.1. Materials

Analytical standards of maleic acid, DL-malic acid, D-(−)-quinic acid, (+)-catechin, (−)-epicatechin, (−)-catechin 3-gallate, L-methionine, L-tryptophan, as well as potassium pyrosulfite (E224), were purchased from Merck (Darmstadt, Germany); gallic acid, quercetin, ABTS (2,2′-azino-bis(3-ethylbenzothiazoline-6-sulfonic acid) diammonium salt and 6-hydroxy-2,5,7,8-tetramethylchromane-2-carboxylic acid (Trolox) from POL-AURA (Zabrze, Poland); tartaric acid from Chempur (,Piekary Slaskie, Poland); D-(+)-xylose and sucrose from SUPELCO (Pennsylvania, PA, USA), and DPPH (2,2-diphenyl-1-picrylhydrazyl from Sigma Aldrich (Saint Louis, MO, USA).

All standards used were of analytical grade (≥99% purity).

The shower gel formulations were made using certified, vegetable-based raw materials which are approved for the production of natural products according to ECOCERS and COSMOS standards: sodium coco-sulfate (Sulfopon 1216 G, BASF, Ludwigshafen, Germany), decyl glucoside (Plantacare 2000, BASF, Ludwigshafen, Germany), cocamidopropyl betaine (Rokamina K30, PCC-Excol, Brzeg Dolny, Poland), benzyl alcohol, benzoic acid, dehydroacetic acid, tocopherol as a preservative (Schülke & Mayr GmbH, Norderstedt, Germany), sodium chloride (POCH, Gliwice, Poland), citric acid (Krakchemia, Kraków, Poland), and distilled water.

### 3.2. Plant Material

All red grapes of the hybrid varieties were cultivated on the Estro Vineyard established in 2019 and grown on a 20 to 30-degree slope and south-west exposure at the foot of the Wrocław Bishops’ Palace ruins at an altitude of 195 m above sea level, in the immediate vicinity of the Jordan stream retention reservoir in Ujazd (Opolskie Voivodeship, Strzelce Opolskie district).

All the grapes were harvested at full maturity (the beginning of October 2021) with entirely black/purplish berries.

On the 10 October 2021, red grapes of the hybrid varieties Regent (Diana × Chambourcin), Léon Millot (Goldriesling × Vitis rupestris × Vitis riparia), Rondo (Zarya Severa × Saint Laurent) were harvested. The grape pomace was destemmed in an OMAC manual grape crusher destemmer (ARES 15 model). After destemmination, and before pressing, the grape pomace was sulfurized with potassium pyrosulfite (E224) at a ratio of 20 g/100 L to stabilize the grape must and to protect it against bacterial and wild yeast growth, as well as against its oxidation. In the next step, grape pomace was pressed in a fruit and wine wooden press to obtain averaged pomace from different varieties (according to a self-developed methodology from the ESTRO Vineyard).

For each grape variety, quality parameters, such as acidity, pH and sugar were measured (Table 6).

The climate in the production area is temperate; the average annual temperature is ca. 9.5 °C; the maximum month temperature is ca. 19.9 °C (July) and the minimum ca. −1.5 °C (January). The average annual rainfall is ca. 750 mm and the humidity ca. 74.4%. The sum of active temperatures (SAT) for the season was ca. 2650 SAT (temperature recorder Elitech rc-5 data logger with a temperature range from −30 to +70 °C and measurement accuracy of ±0.5 °C). For all the varieties the berries were medium-sized spheroids (ca. 15 mm); the bunches were loose with an average bunch weight of approximately 150–200 g.

### 3.3. Extraction Medium Characterisation

#### 3.3.1. Surface Tension Measurements

The surface tension [γ] of the aqueous solutions of the extraction medium was determined at 25 °C using a Krüss K9-Mk1 tensiometer (Krüss, Hamburg, Germany) equipped with a platinum ring. The average values of equilibrium γ were obtained by the measurement repeated three times. The CMC measurements began with a highly concentrated aqueous solution, which was then diluted stepwise by the addition of water. The serial dilutions were prepared from an initial stock solution of the surfactant. The CMC was determined from the graph of surface tension versus surfactant concentration. The breakpoint in the curve was taken as the CMC value [97].

#### 3.3.2. Dynamic Light Scattering (DLS)

Particle size analysis of the extraction medium was performed by dynamic light scattering (DLS), using the Zetasizer Nano ZS Malvern Instrument (Malvern, UK). The sample of decyl glucoside solution was prepared at a concentration of 2% (*w*/*w*). DLS yielded the hydrodynamic diameter (z-average) and the polydispersity index (PDI) as a measure of the particle size distribution. The hydrodynamic diameters (d_h_) were calculated from the Stokes−Einstein equation:d_h_= k_B_ T/3πηD(1)
where k_B_ is the Boltzmann constant, T is the absolute temperature, η is the solvent viscosity, and D is the diffusion coefficient [98]. The autocorrelation functions of the scattered intensity were analyzed by means of the cumulant method to yield the effective diffusion coefficient (D) as a function of the scattered angle. The mean diameter and PDI of the investigated samples were obtained by calculating the average of three measurements, an angle 173°, in disposable polystyrene cuvettes. All the experiments were performed at 25 °C.

### 3.4. Preparation of Micellar Extracts from Grape Pomace Using LCE, as Cosmetic Components Borrowed from the Final Product (Shower Gel)

The micellar extraction process was performed using a mechanical stirrer (CAT, R50D; M. Ziperer GmbH, Ballrechten-Dottingen, Germany). A 2% (*w*/*w*) aqueous solution of surfactant—decyl glucoside was used as the extraction medium. Grape pomace was frozen and grounded with dry ice in a laboratory knife mill (Cutter Mixer R5 Plus, Robot Coupe, Palinges, France). To prepare GPE_10pDG2p to 450 g of extraction medium, 50 g of grounded grape pomace was added and stirred vigorously at 380 rpm. To prepare GPE_20pDG2p, 100 g of grounded grape pomace was added to 400 g of extraction medium. The process was carried out for 3 h at room temperature. The formulation used to prepare the extracts is presented in Table 7.

The obtained extract was filtered under vacuum (Vacuum Pump V-700, Büchi, Flawil, Switzerland) using a non-woven polyester filter (ULESTER 32S (258/231 µm), Silk & Progress, Brněnec, Czech Republic). The filtrate was centrifuged at 7500 rpm for 5 min (Universal 320R centrifuge, Andreas Hettich GmBH &Co, Tuttlingen, Germany) and used directly in further studies.

### 3.5. Determination of Bioactive Compounds by UPLC-ESI–MS/MS

#### 3.5.1. Quantitative UPLC-ESI-MS/MS Analysis of Selected Compounds in Grape Pomace Extracts

The analyses of the selected compounds were performed in independent replicates using four different dilutions of each tested extract in acetonitrile. The extract solutions were filtered through 0.2 µm syringe filters and separated using an ultra-performance liquid chromatography system (UPLC) (Sciex ExionLC AD, AB Sciex, Concord, ON, Canada), equipped with a reverse-phase pre-column and column (Kinetex 3.5 µm XB-C18 100 Å; 100 × 4.6 mm, Phenomenex, Torrance, CA, USA) maintained at 30 °C. The mobile phase consisted of 0.5% (*v*/*v*) aqueous formic acid as solvent A and methanol as solvent B. The gradient elution conditions were as follows: 0.0–20.0 min 15–50% B, 20.0–25.0 min 50% B, 25.0–25.1 min 50–15% B, and 25.1–30.0 min 55% B. The flow rate of the mobile phase was 0.5 mL/min with an injection volume of 10 μL.

The MS detection was performed using a triple quadrupole mass spectrometer (4500 QTRAP, AB Sciex, Concord, ON, Canada), equipped with an electrospray ionization source (ESI) working in positive and negative-scan modes. The parameters of the ionization source were as follows: ion spray voltage, 4500 V (positive-ion mode) and −4500 V (negative-ion mode); source temperature, 650 °C; nebulizing gas, 50 psi; drying gas, 50 psi; and curtain gas, 35 psi. The data were processed using Analyst ver.1.7.2.

The identification of selected compounds was conducted by molecular mass and fragment of anion/cation entries of each compound and confirmed by MS^2^ fragmentation.

For quantification analysis of GPE_10pDG2p and GPE_20pDG2p, the triple quadrupole MS detector was applied in multiple reaction monitoring (MRM) scan mode. The selection of product ions for the individual compounds and the optimal mass analyzer conditions were determined by direct injection of the investigated standard solutions (1 ng/mL) in the mobile phase using an infusion pump. The MS/MS transitions, including declustering potential (DP), entrance potential (EP), collision cell exit potential (CXP), and collision energy (CE), were optimized and are summarized in Appendix A.

The calibration cur ves were generated using peak areas of the most intense MRM transitions of analytical standards for the quantification of 13 detected compounds. The linearity of the detector response for quantified compounds was demonstrated by injection of calibration standards at eight concentration levels, ranging from 0.2 to 40 μg/mL, and for catechin, from 0.02 to 2.5 μg/mL. Standard stock solutions were prepared by accurately weighing and dissolving 10 mg of each standard in 10 mL LC-MS grade methanol to give a concentration of 1000 µg/mL. All dilutions were performed using LC-MS grade methanol. For analysis, 100 µL of grape pomace extracts were dissolved in 900 µL methanol in a chromatographic amber vial.

#### 3.5.2. Qualitative UPLC-ESI-MS/MS Determination of Selected Anthocyanins in Grape Pomace Extracts

For qualitative anthocyanin identification, the same chromatographic parameters as for selected compounds were carried out. Targeted data analysis was performed based on a database built by searching the literature for known grape components [69,74]. The identification was carried out using the molecular mass and MS/MS fragmentation patterns. For a comparison of individual anthocyanins in GPE_10pDG2p and GPE_20pDG2p, the ratio was calculated based on the size of the peak areas.

### 3.6. Determination of Antioxidant Properties

#### 3.6.1. Total Phenolics Content (TPC)

The Folin–Ciocalteu method was used to spectrophotometrically determine total phenolic compounds, as described by Singleton et al. [99]. Briefly, 40 µL of prepared solutions (20 µL of extract and 20 µL of distilled water) was mixed with 200 µL of Folin–Ciocalteu reagent (FC) and 600 µL of 20% Na_2_CO_3_ solution. Then, the solutions were diluted to 4 mL and kept in the dark for 120 min. After this time, the absorbance was measured at 765 nm (Spectrophotometer HP- Hewlett Packard, type: 8452A, Palo Alto, CA, USA) against the blank. The TPC was expressed as milligrams of gallic acid equivalents (GAE) per L extract and was determined as the average from three parallel measurements.

#### 3.6.2. Total Flavonoid Content (TFC)

The total flavonoid content was determined using the colorimetric method of Chang et al. (2002) with some modifications [100]. Briefly, 100 µL of the extract was mixed with 900 µL MeOH, 60 µL of 10% aluminum chloride and 60 µL of 1 M sodium acetate. The mixture was diluted to 4 mL and allowed to stand for 30 min. The absorbance was measured at 420 nm against the blank (mixture of water and reagent). The results were expressed as quercetin equivalents (QE) in milligrams per L and determined as the average of three parallel measurements.

#### 3.6.3. Total Anthocyanin Content (TAC)

The total anthocyanin content of the extracts was estimated by a pH differential method based on the structural changes in chemical forms of anthocyanin [101]. The colored oxonium form predominated at pH 1.0 and the colorless hemiketal form at pH 4.5. Crude extracts were diluted separately with potassium chloride buffer (pH = 1) and sodium acetate buffer (pH = 4.5). The absorbance was measured at 510 nm and 700 nm. The total anthocyanin content was expressed as cyanidin-3-glucoside equivalents (Cyd-3-glu), as in the following equation:TAC = (Abs × MW × DF × 10^3^)/(ε × l)(2)
where: Abs = (A520 nm-A700 nm)pH1.0-(A520 nm-A700 nm)pH4.5; MW—molecular weight Cyd-3-glu, 449.2 g/mol, DF—dilution factor, ε—molar absorptivity, 26,900 L mol^−1^ cm^−1^, and l—cell path length, cm.

#### 3.6.4. Antioxidant Activity (DPPH Test)

The antiradical activity of the extract was evaluated by a modified method reported by Brand-Williams et al. [102]. First, a volume of 1 mL methanol extract (20 µL of an extract with 980 µL of MeOH) was mixed with 3 mL of a methanolic solution of DPPH radical (0.1 mM). The mixture was allowed to stand in the dark for 30 min. The absorbance was measured at 517 nm against the blank (methanol). The antiradical activity of the extract was presented as mg Trolox equivalent (TE)/L of extract.

#### 3.6.5. Antioxidant Activity (ABTS Test)

The antioxidant activity of samples was determined by the method described by Re et al. [103]. ABTS^+^ was obtained in the reaction of 1 mL ABTS [0.01 M 2,2-azino-bis-(3-ethylbenzothiazoline-6-sulfonic acid) diammonium salt] with 1 mL potassium persulfate (0.005 M). The mixture was left to stand in the dark for 20 h to reach optimal absorbance at 734 nm. Then 300 μL of each solution (20 μL extract with 980 µL of distilled water) was mixed with 2 mL of the ABTS^+^ solution, and, after 6 min, the absorbance was measured against a blank (water). The antiradical activity was expressed as mg Trolox equivalent per L of extract.

### 3.7. Preparation of Model Cosmetics (Shower Gels)

Model cosmetics (shower gels) were prepared. All the components used were consistent with EcoCert and COSMOS requirements. The formulation is shown in Table 8.

The product was produced by mixing the ingredients (from item 1 to item 10) at room temperature until a homogeneous product was obtained. In the last step, the viscosity of the formulation was adjusted by addition of sodium chloride.

### 3.8. Determination of the Color Parameters of Grape Pomace Extracts and Cosmetics (Shower Gels) Containing Extracts

The extraction medium, samples of the extracts and cosmetics, with and without grape pomace extracts, were subjected to reflectance color measurements using a Konica Minolta CM-3600 (Konica Minolta, Sensing Singapore Pte Ltd., Tokyo, Japan) with color data software CM-S100w, SpectraMagic NX, ver. 1.07 (Konica Minolta Sensing Europe B.V., Nieuwegein, Netherlands) and light source D65, which represented a daylight phase. Because the human retina has three types of color photoreceptors, three numerical components are necessary and sufficient to describe color. The CIE LAB system (defined by the International Commission on Illumination in 1978), organizes the perception of the three-dimensional color space represented in rectangular coordinates: L*, a*, b*. The L* axis is associated with the lightness of the color: a value of 100 corresponds to white, and a value of 0 to black. The axes a* and b* are associated with the changes of red–green (+a* correlates with red, −a* with green) and yellow–blue (+b* corresponds to yellow, −b* to blue).

The chromaticity coordinates (a*, b*) are represented in polar coordinates by the chroma (C*), the Euclidian distance from the lightness axis and the hue angle (h*), expressed in degrees, starting from the positive a* axis (red) and turning in an anti-clockwise direction.
(3)$$C^{*}=\sqrt{{\left(a^{*}\right)}^{2}+{\left(b^{*}\right)}^{2}}$$
(4)$$h^{o}=\;\text{arctan}\frac{b^{*}}{a^{*}}$$

The differences (DE) between the color of the extraction medium/cosmetic with extract and base extraction medium/cosmetic were calculated by the formula:(5)$$\Delta E_{\text{cosmetic with extract}\;/\text{base cosmetic}\;}^{*}=\sqrt{{\left(\Delta L^{*}\right)}^{2}+{\left(\Delta a^{*}\right)}^{2}+{\left(\Delta b^{*}\right)}^{2}}$$
where ΔL*, Δa*, and Δb* are the mathematical differences between the cosmetics/extraction medium with extracts L*, a*, b* and the base cosmetic/extraction medium L*, a*, b* values.

### 3.9. Rheological Behavior

The viscosity of the model shower gels formulations was measured at 20 °C in triplicate, using a Brookfield rheometer DV2TRV with a small sample adapter and cylindrical spindle SC4 (Brookfield, WI, USA). An aliquot of 8 mL of the sample was used in each test. Different shear rates and shear stresses were applied to the sample, and the resulting rheogram was constructed to determine the rheological behavior

### 3.10. Foam Ability

Foaming property measurements were carried out according to PN-74 C-04801. The measuring cylinder of the Ross–Miles apparatus was used for foam height measurements. Solutions of 10% concentration of the tested preparations were prepared. An aliquot of 50 mL of the solution was poured into the cylinder and a 200 mL portion of the solution was poured into a dropper. The dropper was placed 1 m above the liquid level in the cylinder and the dropper tap was turned on. The height of the foam formed in the cylinder was measured after 1 and 10 min.

### 3.11. Determination of Irritant Potential—Zein Value

In the Zein test procedure, 2 g of protein was solubilized in 40 g of a test sample of the solution of the shower gels: SG_GPE_10pDG2p and SG_GPE_20pDG2p. The amount of the solubilized protein was determined by Kjeldahl analysis, and the result of the Zein value procedure was expressed as mg of the solubilized protein in 100 mL of the sample. The final result was the arithmetic mean of three independent measurements. The test methodology has been described by Wasilewski et al. [87].

### 3.12. Transepidermal Water Loss (TEWL) and Skin Hydration Measurements

Measurements of TEWL, skin hydration, and skin pH were performed with the TEWAmeter TM 300, skin pH-meter PH905 and Corneometer CM 825 probe connected to the MPA adapter (Courage + Khazaka Electronic, Cologne, Germany). The study was conducted on 5 volunteers. Four areas (measuring 2 × 2 cm) were marked on the skin of each volunteer’s forearm. A quantity of 100 mg of each sample was applied in 3 areas. One field (control field) was not treated with any sample. The samples were gently spread over each field and left for 1 min, then rinsed. After 60, 120, 240 and 360 min, the level of hydration, pH, and TEWL were measured. The final result was the arithmetic mean (from each volunteer) of 5 independent measurements (skin hydration and pH), and 20 measurements (TEWL).

### 3.13. Microbiological Stability

Microbiological stability of the extracts and shower gels was evaluated by tests using microcount^®^ duo microbiological testers (Schülke & Mayr GmbH, Norderstedt, Germany). Samples were collected using a sterile swab. The swab was then streaked out onto the agar surface of the slide. Microbiological plates were placed in the tester chamber. The plates were stored at 28 °C, for 3 days for bacterial colony and fungi assays, and for 5 days for yeast and mold testing. After this time, the plates were visually evaluated and the number of microorganisms was determined, based on the test manufacturer’s template.

### 3.14. Stability

Detergent stability was measured with a mechanical loading test. After 24 h of detergent preparation, the samples were subjected to centrifugal force using a Hettich Universal type 320R centrifuge. Each sample was centrifuged at 3000 rpm for 30 min at room temperature.

### 3.15. Statistical Analysis

All UPLC-ESI-MS/MS data were reported as a mean ± standard deviation (SD) with 4 replications for each sample (*n* = 4).

Mean values were compared by ANOVA and a Tukey’s HDS post hoc test.

Calculations were performed using the software package Statistica ver. 10 (StatSoft, Tulsa, OK, USA). A correlation matrix was used to find a significant correlation between the considered variables. Differences were considered significant when the *p*-value was <0.05.

## 4. Conclusions

The cosmetics industry is a growing field, which has increased the demand for new ingredients, preferably from natural sources. For this reason, the use of waste raw materials from wine production as a source of phytochemicals in cosmetic formulations can contribute to the growth of this industry and, consequently, to environmental and economic sustainability. This study provided, for the first time, a comprehensive assessment of the potential use of the loan chemical extraction process in the cosmetics industry.

In our study, an extract obtained from the grape pomace using the loan chemical extraction process contained a high concentration of bioactive compounds. It was shown that this composition exhibited antioxidant activity, and, as a result, can protect the skin against negative environmental influences. The biological activity of compounds from grape extracts favors their valorization as a source of bioactive phytochemicals for use in cosmetics and provides an effective and environmentally friendly alternative for the use of the resulting residues.

The study undertaken also resulted in the development of model shower gels that were designed using the loan chemical extraction technique. The obtained model products showed more benefits for the washed skin; they caused less irritation and had a lower influence on its drying (in comparison with analogous shower gels not containing extracts). Additionally, the products had natural color in red shades, which was well perceived by the consumer. It was observed that the addition of extracts did not have a significant influence on the parameters related to functionality (e.g., foaming properties, viscosity). In conclusion, it can be stated that the production of shower gels with the use of the loan chemical extraction technique significantly improved the safety of this type of preparation and, thus, significantly improved the quality of products. In addition, the interesting possibility of utilizing waste generated in the production of wine has been demonstrated.

## Figures and Tables

**Figure 1 molecules-27-02444-f001:**
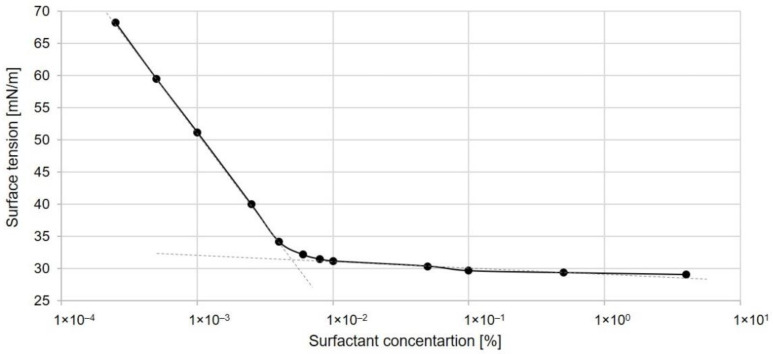
Variation in surface tension as a function of surfactant concentration.

**Figure 2 molecules-27-02444-f002:**
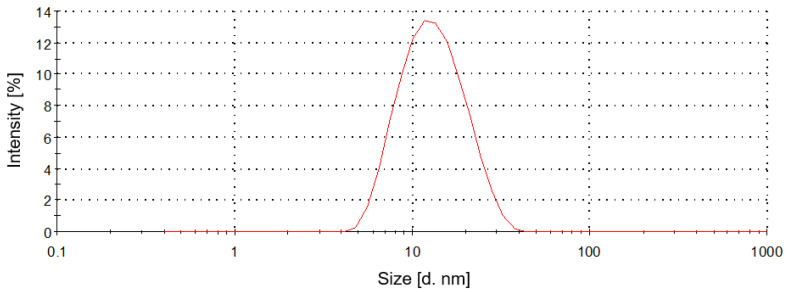
Particle size distribution curves of the extraction medium.

**Figure 3 molecules-27-02444-f003:**
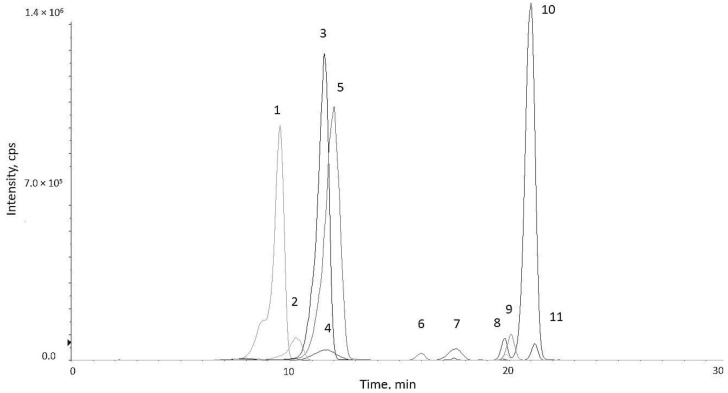
UPLC-ESI-MS/MS extracted ion chromatograms (XICs) in positive-ion mode for GPE_10pDG2p. The numbering of the peaks is by the retention time. Peak assignment is reported in Table 2.

**Figure 4 molecules-27-02444-f004:**
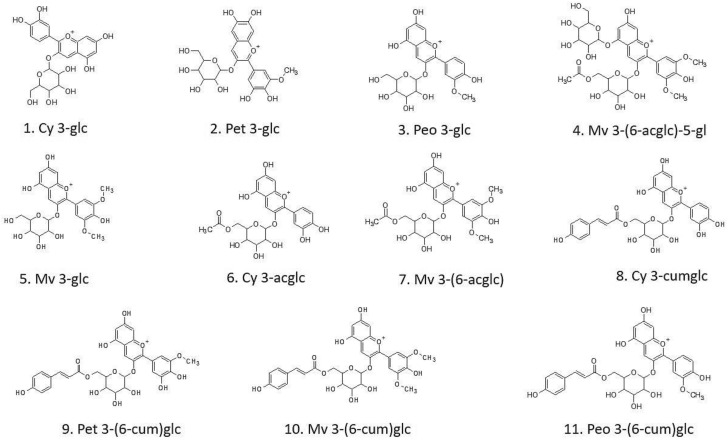
Characterization of identified anthocyanins in the order of elution time.

**Figure 5 molecules-27-02444-f005:**
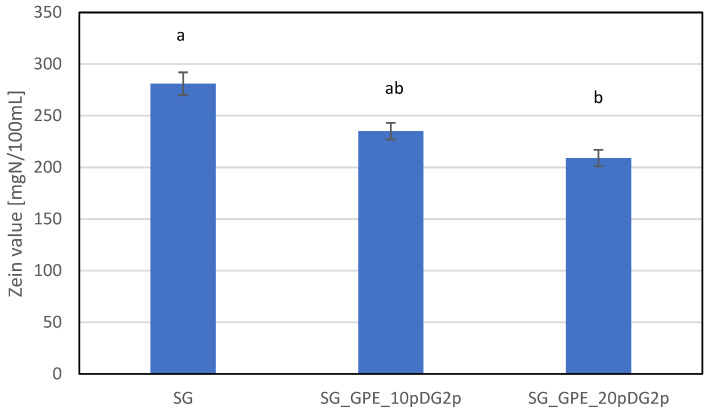
The irritant potential of model bodywash gels. The superscripts a and b denote significant (*p* < 0.05) differences between the test extracts.

**Figure 6 molecules-27-02444-f006:**
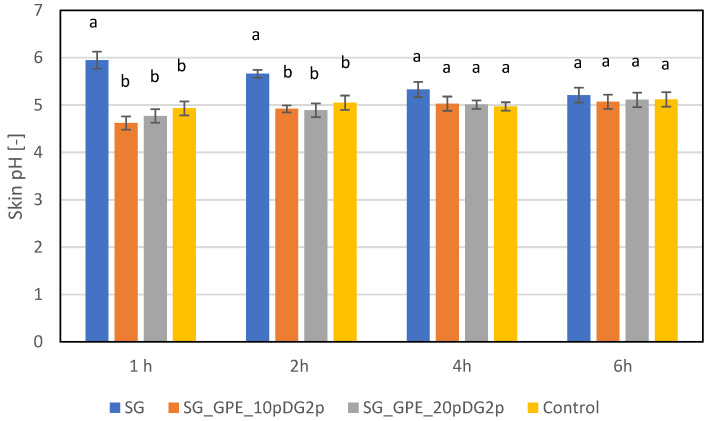
Influence of plant extracts on skin pH. The superscripts a and b denote significant (*p* < 0.05) differences between the test extracts within the individual test times.

**Figure 7 molecules-27-02444-f007:**
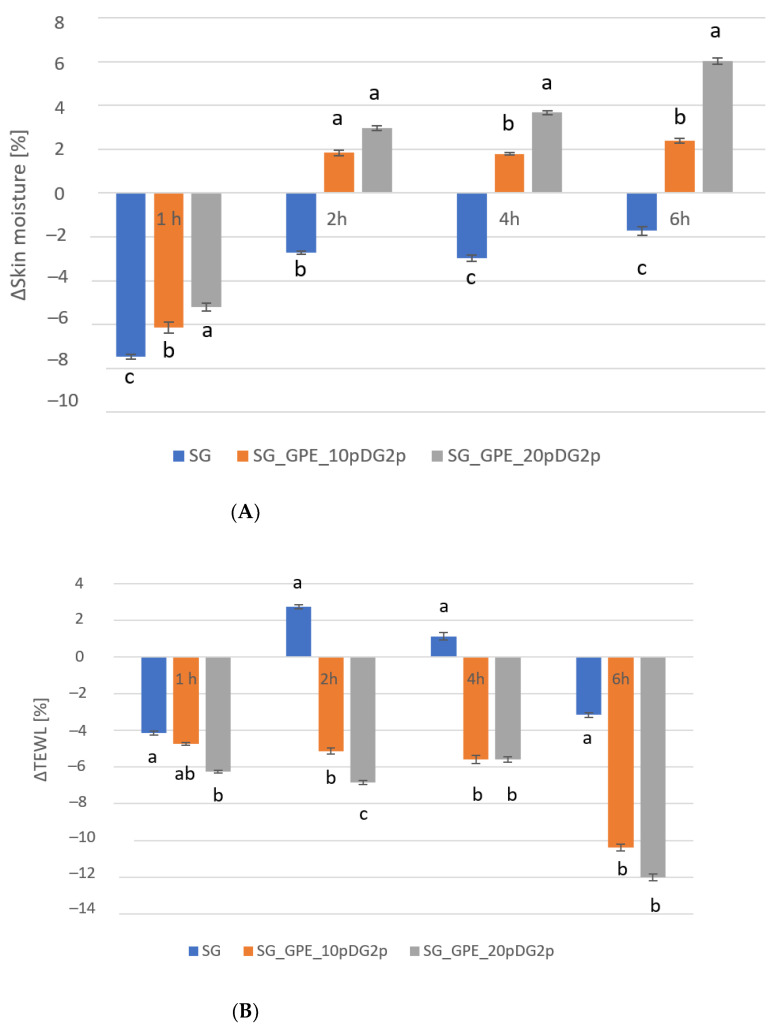
Influence of plant extracts on skin hydration (**A**) and TEWL (**B**). The superscripts a, b and c denote significant (*p* < 0.05) differences between the test extracts within the individual test times.

**Figure 8 molecules-27-02444-f008:**
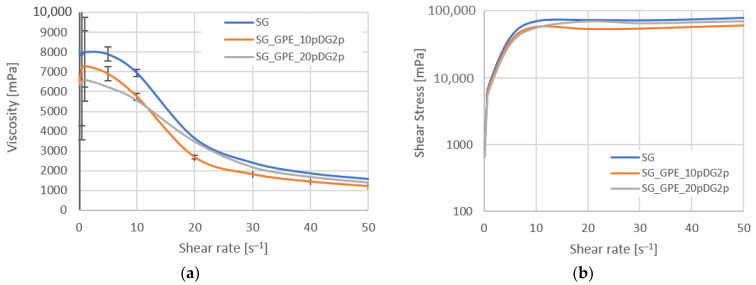
Rheogram of detergents; (**a**) Viscosity vs. shear rate; (**b**) Shear stress vs. shear rate.

**Figure 9 molecules-27-02444-f009:**
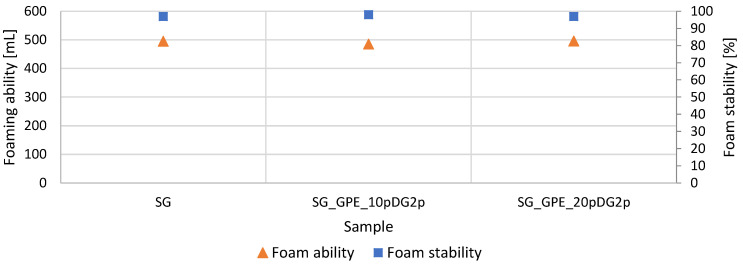
Foam properties.

**Figure 10 molecules-27-02444-f010:**
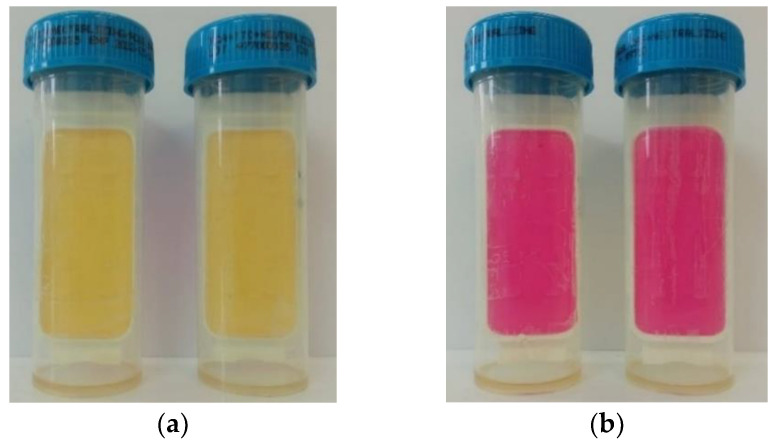
Example plates for microbiological stability testing: (**a**) microcount^®^ duo plate for total bacterial colonies count, (**b**) microcount^®^ duo plates for yeasts, fungi, and molds culture.

**Table 1 molecules-27-02444-t001:** UPLC-ESI-MS/MS quantification of the detected compounds in tested grape pomace extracts mean values ± standard deviation (*n* = 4).

Compound		Quantification/ Confirmation Transition	Family	GPE_10pDG2p [mg/L]	GPE_20pDG2p [mg/L]
1	Tartaric acid	148.9 > 87.0 148.9 > 73.0	organic acids	1104 ± 11 ^b^	2817 ± 17 ^a^
2	Maleic acid	114.9 > 70.9 114.9 > 45.0	organic acids	4.20 ± 0.80 ^a^	5.15 ± 0.48 ^a^
3	DL-malic acid	132.9 > 114.9 132.9 > 71.0	organic acids	848 ± 11 ^b^	1147 ± 17 ^a^
Sum of organic acids				1956 ^b^	3969 ^a^
4	Gallic acid	168.9 > 124.8 168.9 > 78.9	phenolic acids	1.87 ± 0.19 ^b^	4.91 ± 0.33 ^a^
5	D-(−)-quinic acid	190.9 > 84.9 190.9 > 93.0	phenolic acids	4.88 ± 0.26 ^b^	10.86 ± 0.17 ^a^
6	Quercetin	300.9 > 151.0 300.9 > 179.0	flavonols	0.26 ± 0.08 ^b^	0.27 ± 0.03 ^a^
7	(+)-Catechin	290.9 > 139.0 290.9 > 123.0	flavanols	5.76 ± 0.06 ^a^	5.36 ± 0.1 ^a^
8	(−)-Epicatechin	290.9 > 139.0 290.9 > 123.0	flavanols	4.40 ± 0.22 ^a^	4.64 ± 0.10 ^a^
9	(−)-Catechin 3-gallate	306.9 > 288.8 306.9 > 163.0	flavanols	2.67 ± 0.10 ^b^	4.23 ± 0.17 ^a^
Sum of phenolic compounds				19.9 ^b^	30.3 ^a^
10	L-methionine	150.0 > 103.9 150.0 > 132.9	amino acids	2.78 ± 0.08 ^b^	6.10 ± 0.77 ^a^
11	L-tryptophan	205.1 > 188.0 205.1 > 145.9	amino acids	3.05 ± 0.25 ^b^	6.96 ± 0.16 ^a^
Sum of amino acids				5.8 ^b^	13.1 ^a^
12	D-(+)-xylose	149.9 > 104.0 149.9 > 89.9	sugars	776 ± 12 ^b^	1417 ± 39 ^a^
13	Sucrose	340.9 > 179.0 340.9 > 88.9	sugars	2.77 ± 0.13 ^b^	3.50 ± 0.13 ^a^
Sum of sugars				779 ^b^	1420 ^a^
TOTAL				2760 ^b^	5432 ^a^

The superscripts a and b denote significant (*p* < 0.05) differences between the two concentrations within each compound identified.

**Table 2 molecules-27-02444-t002:** Anthocyanin compounds identified using UPLC-ESI-MS/MS in positive-ion mode in GPE_10pDG2p and GPE_20pDG2p.

Peak	Retention Time [min]	Identification	Molecular Formula	Molar Mass [Da]	Precursor Ion *m*/*z*	Main Product Ions MS^2^ [*m*/*z*]	CE [V]	GPE_20pDG2p/GPE_10pDG2p Ratio
1	9.5	Cyanidin 3-glucoside (Cy 3-glc)	C_21_H_21_O_11_^+^	449	449 [M + H]^+^	287 [M-C_6_H_11_O_5_]^+^ 315 [M-C_5_H_10_O_4_]^+^	24	1.3 ± 0.1 ^b^
2	10.3	Petunidin 3-glucoside (Pet 3-glc)	C_22_H_23_O_12_^+^	478	479 [M + H]^+^	317 [M-C_6_H_11_O_5_]^+^ 302 [M-C_6_H_11_O_4_]^+^	10	1.9 ± 0.2 ^a^
3	11.6	Peonidin 3-glucoside (Peo 3-glc)	C_22_H_23_O_11_^+^	462	463 [M + H]^+^	301 [M-C_6_H_11_O_5_]^+^ 201 [C_9_H_5_O_4_]^+^	20	1.1 ± 0.1 ^b^
4	11.7	Malvidin-3-(6”-acetoyl)glucoside-5-glucoside	C_31_H_37_O_18_^+^	696	697 [M + H]^+^	535 [M-C_6_H_10_O_5_]^+^ 331 [M-C_14_H_24_O_11_^]+^	34	1.8 ± 0.1 ^a^
5	12.1	Malvidin 3-glucoside (Mv 3-glc)	C_23_H_25_O_12_^+^	492	493 [M + H]^+^	331 [M-C_6_H_11_O_5_] ^+^315 [M-C_5_H_10_O_4_]^+^	15	1.2 ± 0.2 ^b^
6	16.0	Cyanidin 3-(acetylglucoside) (Cy 3-acglc)	C_23_H_23_O_12_^+^	490	491 [M + H]^+^	287 [M-C_8_H_13_O_6_]^+^ 163 [M-C_17_H_13_O_7_]^+^	18	1.2 ± 0.1 ^b^
7	17.6	Malvidin 3-(6”acetyl) glucoside (Mv 3-(6-acglc))	C_25_H_27_O_13_^+^	534	535 [M + H]^+^	331 [M-C_8_H_13_O_6_]^+^ 315 [M-C_8_H_13_O_7_]^+^	12	1.2 ± 0.1 ^b^
8	19.9	Cyanidin 3-O-p-coumarylglucoside (Cy 3-cumglc)	C_30_H_27_O_13_^+^	594	595 [M + H]^+^	287 [M-C_15_H_17_O_7_]^+^ 415 [M-C_9_H_8_O_4_]^+^	19	1.0 ± 0.1 ^b^
9	20.1	Petunidin 3-(6”-cumaroyl)-glucoside (Pet 3-(6-cum)glc)	C_31_H_29_O_14_^+^	624	625 [M + H]^+^	317 [M-C_15_H_17_O_7_]^+^ 301 [M-C_15_H_18_O_8_]^+^	23	1.4 ± 0.1 ^ab^
10	21.0	Malvidin 3-(6”-cumaroyl)-glucoside (Mv 3-(6-cum)glc)	C_32_H_31_O_14_^+^	638	639 [M + H]^+^	331 [M-C_15_H_17_O_7_]^+^ 447 [M-C_6_H_10_O_5_]^+^	35	1.1 ± 0.1 ^b^
11	21.0	Peonidin 3-(6”-cumaroyl)-glucoside (Peo 3-(6-cum)glc)	C_31_H_29_O_13_^+^	608	609 [M + H]^+^	301 [M-C_15_H_17_O_7_]^+^ 492 [M-C_9_H_8_O_4_]^+^	20	0.9 ± 0.1 ^b^

Ratio as a mean value ± standard deviation (*n* = 4). The superscripts a and b denote significant (*p* < 0.05) differences between ratios of all the identified compounds.

**Table 3 molecules-27-02444-t003:** Antioxidant capacity (DPPH, ABTS), total phenolic (TPC), flavonoid (TFC), and anthocyanin (TAC) content in grape extracts.

	TPC [mg GAE/L]	TFC [mg QE/L]	TAC [mg Cyd-3-glu/L]	DPPH [mg TE/L]	ABTS [mg TE/L]
	TPC ± SD	TFC ± SD	TAC ± SD	DPPH ± SD	ABTS ± SD
GP_10pDG2p	543.1 ^b^ ± 12.7	139.2 ^b^ ± 11.2	236.7 ^b^ ± 18.6	815.4 ^b^ ± 10.3	743.6 ^b^ ± 5.2
GP_20pDG2p	709.1 ^a^ ± 6.7	177.6 ^a^ ± 5.6	395.7 ^a^ ± 6.9	1126.6 ^a^ ± 24.7	954.3 ^a^ ± 10.2

Each value represents the average from three parallel measurements. Values are expressed as mean ± SD. The superscripts a, b denote significant (*p* < 0.05) differences between the obtained results.

**Table 4 molecules-27-02444-t004:** Spectrophotometric data of the GPE_10pDG2p and GPE_10pDG2p obtained by D65.

	L*	a*	b*	C*	ho	ΔE
DG2p	27.16 ± 0.13 ^c^	−0.55 ± 0.05 ^c^	0.18 ± 0.04 ^c^	0.6 ± 0.1 ^c^	−18.1 ± 0.3 ^c^	-
GPE_10pDG2p	5.25 ± 0.05 ^a^	20.53 ± 0.07 ^a^	8.38 ± 0.07 ^a^	22.2 ± 0.2 ^a^	22.2 ± 0.3 ^a^	31.5 ± 0.4 ^a^
GPE_20pDG2p	2.55 ± 0.03 ^b^	15.62 ± 0.06 ^b^	4.44 ± 0.05 ^b^	16.2 ± 0.2 ^b^	15.7 ± 0.2 ^b^	29.7 ± 0.4 ^b^

ΔE compared to extraction medium DG. Values are means of five replicate determinations ± standard deviation (*n* = 5). The superscripts a, b and c denote significant (*p* < 0.05) differences within a given color parameter between extracts.

**Table 5 molecules-27-02444-t005:** Spectrophotometric data of the cosmetics with GPE_10pDG2p and GPE_20pDG2p obtained by D65.

	L*	a*	b*	C*	ho	ΔE
SG	30.72 ± 0.12 ^c^	−0.06 ± 0.05 ^c^	0.55 ± 0.03 ^c^	0.6 ± 0.1 ^c^	−83.8 ± 0.3 ^c^	-
SG_GPE_10pDG2p	7.75 ± 0.06 ^a^	30.45 ± 0.06 ^a^	11.87 ± 0.05 ^a^	32.7 ± 0.2 ^a^	21.3 ± 0.2 ^a^	39.8 ± 0.3 ^a^
SG_GPE_20pDG2p	4.16 ± 0.04 ^b^	23.17 ± 0.05 ^b^	7.16 ± 0.04 ^b^	24.3 ± 0.2 ^b^	17.2 ± 0.2 ^b^	35.9 ± 0.4 ^b^

ΔE extracts compared to SG. Values are means of five replicate determinations ± standard deviation (n = 5). The superscripts a, b and c denote significant (*p* < 0.05) differences within a given color parameter between cosmetics.

**Table 6 molecules-27-02444-t006:** Quality parameters of harvested grapes (according to the Compendium of International Methods Of Analysis—OIV) [96].

	Acidity [g/L]					pH	Sugar [°Bx]
	Tartaric Acid	Malic Acid	Citric Acid	Acetic Acid	Sulphuric Acid		
Regent	6.75	6.03	5.76	5.40	4.41	3.46	20
Léon Millot	8.80	7.86	7.51	7.04	5.75	3.43	21
Rondo	9.15	8.17	7.81	7.32	5.98	3.23	20

**Table 7 molecules-27-02444-t007:** The formulation used to prepare extracts.

	Ingredient (INCI Name)	GPE_10pDG2p	GPE_20pDG2p
1	Decyl glucoside	2	2
2	Benzyl alcohol, benzoic acid, dehydroacetic acid, tocopherol	0.5	0.5
3	Aqua	87.5	77.5
4	Grape pomace	10	20

**Table 8 molecules-27-02444-t008:** Formulation of a model cosmetic (shower gels).

	Ingredient (INCI Name)	SG_GPE_10pDG2p [%]	SG_GPE_20pDG2p [%]
1	Sodium coco-sulfate	4.5	4.5
2	Aqua	to 100	to 100
3	GPE_10pDG2p	65	-
4	GPE_20pDG2p	-	65
5	Decyl glucoside	3.8	3.8
6	Benzyl alcohol, benzoic acid, dehydroacetic acid, tocopherol	0.5	0.5
7	Citric acid	to pH 5.5	to pH 5.5
8	Parfum	0.5	0.5
9	Cocamidopropyl betaine	2	2
10	Sodium chloride	1	1

## Data Availability

The data presented in this study are available in Appendix A.

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
