# Peer review of "Sustainable Green Processing of Grape Pomace Using Micellar Extraction for the Production of Value-Added Hygiene Cosmetics"

_molecules, 2022, doi:10.3390/molecules27082444_

Round 1

Reviewer 1 Report

The study by Wasilewski et al. evaluated the use of grape pomace, a waste product of wine production, as constituent of cosmetic using the Loan Chemical Extraction method and further tested shower gels produced for functionality properties and effect on the skin, and concluded that the product contained organic acids, phenolics, amino acids and sugars, and that it could be regarded safe when used as a cosmetic agent. While the concept of the study looks good, a number of concerns bothering on technicality abound that need to be addressed. The specific reasons for my recommendation are:

MINOR COMMENTS

  1. The "Introduction" section is unnecessarily too long and needs to be concise and focused.
  2. Several statement in the "Materials and Methods" section needs to be referenced. e.g.,  Lines 559 - 567, Lines 577 - 635 etc.
  3. Correct the inconsistencies in section 3.1 regarding the suppliers where some have details of the companies and others not.
  4. Avoid repetition of 'Methods' and 'Goal' of the study in the 'Results and Discussion' section. This has contributed to the increased length of the manuscript.
  5. Please correct the errors in referencing in some parts of the manuscript. e.g., Lines 301 - 305, Line 647 etc.

MAJOR COMMENTS

  1. It is strange to see the compounds presented in Table 1 as the only set of constituents in the extract and why untargeted UPLC was not performed. This could have possibly afforded more constituents than what has been reported here.
  2. What is [-] representing in Figure 6? And it would have been expected to see superscripts denoting the level of statistically significant variations between the measured pH at different periods. This is equally applicable to Table 7.
  3. In Section 2.5.6., it was stated that the gels were stable without any evidence or reference to parameters supporting this. This is not acceptable and must be addressed. This is the case with some of the other parameters as well.

Reviewer 2 Report

The authors did a good work from an experimental point of view, and I recommend the article for publication after some minor revisions.

More specific:

L57: I suggest further research in this paragraph on deep eutectic solvents (DES).

L301 & 305: Error! Reference source not found.

L575: Where is the grapes pomace extraction methodology? How do you prepare grape pomace extracts?

Reviewer 3 Report

The authors performed impeccable work in experimental as well as in writing.

I just have some very minor indications, for me is an approval. Congratulations to the authors

Best regards

Line 18 This study concerned the evaluation of the possibility of using grape pomace, Please revise English in all the text

Line 36-37 How is it that there could be surfactants “which are sustainable, cheap but can be harmful to the consumer” to be sustainable it would need an LCA indicating sustainability

Line 220 which which repeated

Line 301-305 [Error! Reference source not 301 found.]

Line 373: To determining the effect of the analyzed “For determining”

Figure 6. Would you have  better quality of the figure please, it looks very pixelated
